# Incorporation of Lutein on Layered Double Hydroxide for Improving the Environmental Stability

**DOI:** 10.3390/molecules25051231

**Published:** 2020-03-09

**Authors:** Shue Li, Bin Mu, Wenkai Dong, Oing Liang, Shijun Shao, Aiqin Wang

**Affiliations:** 1Key Laboratory of Clay Mineral Applied Research of Gansu Province, Center of Eco-Materials and Green Chemistry, Lanzhou Institute of Chemical Physics, Chinese Academy of Sciences, Lanzhou 730000, China; seli17@licp.cas.cn (S.L.); dongwenkai5017@163.com (W.D.); liangqing@licp.cas.cn (O.L.); sjshao@licp.cas.cn (S.S.); 2Center of Materials Science and Optoelectronics Engineering, University of Chinese Academy of Sciences, Beijing 100049, China; 3Center of Xuyi Palygorskite Applied Technology, Lanzhou Institute of Chemical Physics, Chinese Academy of Sciences, Xuyi 211700, China

**Keywords:** lutein, layered double hydroxide, composites, environmental stability

## Abstract

To overcome the poor stability of natural lutein to environmental factors, layered double hydroxide was incorporated by a green mechanical grinding process. The influences of external factors (chemical reagents, heating and light) on the stability of lutein before and after being loaded were evaluated. The results confirmed that lutein was mainly adsorbed on the surface of layered double hydroxide (LDH) via the chemical interaction. Compared with pure lutein, the thermal decomposition of lutein/LDH was improved from 100 °C to 300 °C, and the retention ratio of lutein was increased by about 8.64% and 21.47% after 96 h of light exposure and accelerated degradation, respectively. It is expected that the stable lutein/LDH composites may constitutean additive in animal feed.

## 1. Introduction

Carotenoids are a group of natural dyes that include three main colors of red, yellow, and orange, which are determined by the specific conjugated double-bond structure of molecules [1,2]. The carotenoid group can be classified into xanthophylls and carotenes depending on whether it has oxygen-containing functional groups in six-membered rings at either end of a long hydrocarbon chain [3,4,5]. Among them, lutein is a member of xanthophylls containing oxygen atoms (Appendix A), this natural oil-soluble food pigmentis widely distributed in egg yolk, common fruits and dark green leafy vegetables [6,7,8]. The natural lutein pigment has many excellent properties, such as the antioxidant properties possessed by most natural pigments. In addition, it also has great human health benefits like eye protection, prevention of various diseases, reduction of skin damage caused by UV, and so on [9,10,11]. Lutein detected in humans and animals is derived from plants, egg yolk, and processed foods containing lutein [12]. However, the conjugated structure of lutein is sensitive to environmental factors such as light, oxygen, pH, and heat, which limits its application in food processing and feed industries [7,8,13]. Therefore, it is crucial to develop the feasible strategies to enhance the stability and availability of lutein in practice.

At present, great efforts arefocused on the encapsulation of lutein using inclusion complexes, cornstarch granules, whey protein nano-emulsion, and lipid nanocarriers, etc. to improve the stability and bioavailability of lutein [9,14,15,16,17]. The combination of oligosaccharides and polysaccharides molecules with carotenoids lutein and zeaxanthin resulted in the formation of supramolecular complexes, greatly improving in chemical stability, photostability, and oxidation stability of xanthophylls [18,19,20]. Furthermore, Jiao et al. successfully designed the novel nanocarriers based on zein-derived peptide nanoparticles, which noticeably enhanced the solubility and stability of free lutein [5]. In another study, Yang et al. presented a core-shell system to encapsulate lutein using ferritin and chitosan [21]. The storage-, photo-, and heat-stability of lutein in ferritin-chitosan-lutein nanocomposites were greatly improved in comparison to pure lutein.

Compared with the encapsulation method, the loading of lutein is more popular using inorganic carriers, especially layered double hydroxide (LDH) due to the unique intercalated structures. LDH, generally defined as [M^II^_1−x_M^III^_x_(OH)_2_]^x+^[(A^n−^)_x/n_·mH_2_O]^x−^, is a well-known anionic hydrotalcite-like compounds consisted of the positively charged hydroxide layers because of the partial substitution of divalent metal ions by trivalent ones, and the net positive charge on the layerswas neutralized by the interlayer negative anion species and water molecules [22,23,24,25]. Due to its advantages of green, non-toxic, low-cost, and unique intercalated structures, LDH has attracted considerable interest for designing organic/inorganic hybrid materials [26]. Liu et al. and Li et al. obtained two kinds of hybrid compounds by combining azoic dyes with Mg/Al-CO_3_-LDH using the ion exchange method. The results showed that adsorption was more beneficial to improve the stability and the blue emission of dyes than intercalation due to the strong chemical adsorption [27,28]. Moreover, incorporation of Acid Red 114 into Zn/Al-NO_3_-LDHs also significantly contributed to the enhancement of the thermal stability and the light fastness [29]. Furthermore, LDH also played an important role in improving the performances of natural dyes besides synthetic organic ones. For example, the photostable hybrid materials were successfully prepared through inserting several natural dyes, namely β-carotene, annatto, carmine, and carthamus yellow, into hydrotalcite [30,31]. Thereby, it was found that the interlayer polarity of hydrotalcite had a certain influence on the stability of the obtained pigments. More importantly, the multiple interactions between the organic guests and the hydroxide layer were in favor of increasing the stability of dyes including electrostatic and van der Waal’s forces, complexation, hydrogenbonds [31,32,33]. Recently, the mechanochemical method was also employed to prepare LDH and LDH-based intercalation compounds [34,35,36]. However, the synthesis of LDH using metal ions derived from wastewater during clay minerals processing was scarcely reported.

In this study, Mg-Al-Fe LDH was prepared for loading lutein by a facile grinding process using metal ions derived from spent liquor, which was produced during controllable acid-leaching palygorskite for whitening. The characteristics of the chemical resistance, thermal, light and storage stability of lutein/LDH composites were investigated. The possible reason of the improvement in the stability of lutein was also explored. As a result, lutein/LDH composites may be used as a potential feed additive for the source of lutein uptake in animals, and it also provides a feasible approach to realize the sustainable application of spent liquor during the processing of clay minerals.

## 2. Results and Discussion

### 2.1. Characterization of Lutein/LDH Composites

A facile grinding method was successfully used to synthesize lutein-loaded layered double hydroxide composites (lutein/LDH). The possible mechanism of the lutein dyes and LDH is illustrated in Figure 1. The morphologies of LDH and lutein/LDH are shown in Figure 2a. It was clear that the raw LDH presented a typical lamellar structure. After incorporation of lutein, the morphology of the products was similar with that of raw LDH.

The X-ray diffraction (XRD) pattern showed the diffraction characteristics of LDH (Figure 2b). The sharp and narrow diffractions of (003) and (006) planes illustrated that synthesized LDH had a well-defined crystalline structure with orderly stacking of layers [37,38]. The diffraction peak at 2θ of 29.54° was present inXRD patterns of LDH and lutein/LDH composites, which might be related to the formation of CaCO_3_ impurity during the preparation of LDH [39]. With the introduction of lutein, the value of *d_003_* slightly increased from 7.60 Å to 7.64 Å. This demonstrated that lutein molecules did not completely embed into the interlayer of LDH because of the definite three-dimensional structure of lutein. Furthermore, it was clearly established that the strong affinity of CO_3_^2−^ to metal ions of LDH resulted in the absence of anion exchange reactions between lutein and LDH [40,41,42]. However, no obvious changes were observed from non-basal (009), (110) and (113) reflection after incorporation of natural lutein molecules, indicating that the LDH crystals were preserved during the preparation of composites [43]. Therefore, lutein might be mainly adsorbed on the surfaces of LDH or partially intercalated into interlayers of LDH. In addition, it was worth noticing that the lutein/LDH hybrid composites had no new diffraction peaks compared with LDH. It could be explained by the conversion of pristine lutein with high crystallinity to amorphous phase after grinding [44,45].

Fourier transform infrared spectroscopy (FTIR) spectra of LDH, lutein and lutein/LDH composites were provided in Figure 2c. The broad band at around 3455 cm^−1^ could be assigned to the OH-stretching mode of interlayer water molecules and hydroxide groups of layers [46,47]. The peak at 1627 cm^−1^ corresponded to the bending mode of interlayer water. The three absorption bands were observed at 1360, 855, and 673 cm^−1^, which could be ascribed to the asymmetric and symmetric stretch of interlayer carbonates [48,49]. The characteristic bands observed at 983, 780, 554, and 449 cm^−1^ were attributed to the translation, deformation, and vibration modes of the metal hydroxide sheets, respectively [46,49,50,51,52]. The spectrum indicated that the OH stretch in pristine lutein pigments appeared at 3421 cm^−1^. A series of characteristic bands at 2854, 1363, 1125, and 1041 cm^−1^ belonged to the stretching vibrations of CH_3_ and CH_2_ groups, the split of dimethyl group, the C-O stretch, and C-H groups, respectively [53,54]. The -C=C- vibration peak was found at 1610 cm^−1^ [55]. In addition, the two peaks at 963 and 831 cm^−1^ were ascribed to the out of plane bending of -CH=CH- group of lutein [56,57]. Compared with the spectra of lutein and LDH, it was clearly that new characteristic peaks also appeared at 2856, 1609 and 1121 cm^−1^ due to the loading of lutein besides the typical characteristic peaks of LDH. It indicated that lutein was successfully loaded onto LDH without damaging the properties of natural lutein and LDH. Furthermore, the characteristic bands of Al-OH of LDH shifted from 983 cm^−1^ to 976 cm^−1^ (Appendix A), which might correspond to the chemical interaction between lutein and the hydroxide layers of LDH-like H-bonding [58].

The specific surface area (*S_BET_*), the total pore volume (*V_total_*) and average pore width of LDH and lutein/LDH hybrid composites were listed in Table 1, respectively. The average pore size of composites slightly increased compared with that of LDH. By contrast, *S_BET_* and *V_total_* of LDH in composites were significantly reduced by 23.26% and 16.09%, respectively. The above changes of structural parameters before and after the loading of lutein could be attributed to the surface adsorption orpartial intercalation of lutein on LDH [59]. In addition, it is well-known that the hydroxide layer of LDH is positively charged because of the partial substitution of divalent metal ions by trivalent ones, while carbonate negative anions are intercalated into the interlayers for balancing the charge. Thus, the zeta potential of LDH was 3.75 mV, and the zeta potential of LDH decreased after the incorporation of lutein. As a result, there might be electrostatic interactions between natural guests and inorganic hosts.

### 2.2. Environmental Stability of Lutein/LDH Composites

#### 2.2.1. Chemical Stability

Compared with the color parameters of pure lutein dyes, the lightness (*L**) of the lutein/LDH composites were enhanced, but the value of *a**(negative values for green and positive values for red) decreased obviously after addition of LDH (Table 2). The lutein/LDH composites were immersed into 0.1 M NaOH, acetone, ethanol and ethyl acetate solution for 24 h to evaluate the chemicalresistance of samples, respectively. Although the *L** and *a** values of lutein/LDH samples were similar with that of raw lutein/LDH samples after being immersed into acetone, ethanol, and ethyl acetate, respectively, the *b** values (negative-blue/positive-yellow) of above samples were low, and reached 30.88, 29.23 and 31.05, respectively. Obviously, lutein/LDH composites presented better chemical stability against 0.1 M NaOH than three solvents since the *L**, *a**and *b** values of alkali-treated samples had little change compared with that of raw lutein/LDH, especially the values of *L** and *a**. This was in line with the colors of the supernatants obtained after being treated using four solvents (Appendix A). The colors of the supernatants eluted from 0.1 M NaOH was lighter than that of the supernatants eluted from other solvents. This suggested that the lutein/LDH samples presented the optimum stability in alkaline solution. In general, lutein molecules were effectively protected by LDH host from chemical agents, especially alkaline solution.

#### 2.2.2. Thermal Stability

The thermal stability of lutein/LDH composites was evaluated by comparing with the degradation temperature of lutein presented in the thermogravimetric analysis and differential scanning calorimetry (TGA-DSC) curves. Different from most previous reports, the LDH decomposition was complete below 600 °C including two characteristic stages of mass loss [60,61,62], there was three-step mass loss of prime LDH from room temperature to 800 °C corresponding to four endothermic peaks. As presented in Figure 3a, the mass loss of the first stage between 70 and 220 °C could be ascribed to the removal of physically adsorbed and interlayer H_2_O. The second stage in the range of 250–500 °C presented two endothermic peaks located at 308 and 387 °C, which was due to the decomposition of interlayer carbonate anions and the dehydroxylation of LDH layer resulting in collapsing the structure [61,62]. The high temperature exothermic peak appeared at 695 °C corresponding the mass loss occurred between 650 and 710 °C, which was related to the combustion of the residue [59]. The TGA curve of lutein/LDH composites exhibited a trend similar to that of prime LDH, which was completely different from pure lutein (Figure 3b). Compared with the TGA curves of pure lutein (100 °C), the thermal decomposition temperature of lutein loaded on LDH increased to around 300 °C. This revealed that the introduction of LDH significantly improved the thermal stability of natural pigments.

As illustrated in Figure 4, the intensities of the characteristic peaks of pure lutein decreased significantly after being heated at 120 °C due to the degradation of molecules. On the contrary, the characteristic peaks of lutein in composites were observed at 2854, 1610, and 1125 cm^−1^, indicating no obvious changes. Combined with the TGA analysis, this suggested that LDH exhibited a positive effect on improving the thermal stability of lutein.

#### 2.2.3. Photostability and Storage Stability

As shown in Figure 5a, light exposure was used to determine the photostability of composites. Obviously, pure lutein was very sensitive to light. The retention ratio of pure lutein rapidly decreased to 73.27% after being exposed for 8 h, which was about 10% lower than that of composites. After being exposed for 96 h, the retention ratio of pure lutein was just 26.28%, whereas that of lutein loaded on LDH was 34.92%. Compared with the composites of anionic perylene dye and LDHs reported by Bauer et al. the photostability of lutein was significantly improved after incorporation of LDH [63].

Both temperature and light could accelerate degradation of lutein due to many special conjugated double bonds, resulting in a significant reduction in retention of lutein that limited its storage and application for long periods of time [13,64]. As canbe seen from Figure 5b, pure lutein had short-term storage stability at 55 °C in an accelerated degradation experiment. However, the degradation rate of composites was obviously lower than that of pure lutein as the storage time increased. The retention rate of lutein loaded on LDH was about 21.47%, which was higher than that of pure lutein after 96 h after accelerated degradation. In addition, it should be noted that lutein storedat 55 °C for 24 h was equivalent to eight days at 25 °C in this accelerated degradation experiment. Therefore, it could be concluded that the obtained lutein/LDH composites exhibited excellent light resistance and storage stability due to the shielding and protecting effect of LDH. It facilitated the long-term storage and transportation of products containing natural lutein pigments.

## 3. Materials and Methods

### 3.1. Materials

Lutein was extracted and purified from marigold flowers according to the improved literature procedures [65]. In brief, 100 g of marigold mill was homogenized in a blender with 1000 mL of 95% ethanol and 100 mL of 10% ethanolic KOH at 60~65 °C for 2 h. The mixture was filtered and then the filtrate was neutralized with an aqueous solution of H_3_PO_4_ until the pH dropped to 7.0. The solvents were evaporated under reduced pressure at 55 °C and the residues were stirred at room temperature with 250 mL of 30% ethanol, then filtered and the filter cake was washed with 30% ethanol, and dried in vacuum at room temperature to give lutein as yellow crystals (1.8 g, 75% pure lutein). Lutein was recrystallized with mixed solvent of tetrahydrofuran and water (*v/v*, 1:4) to raise the purity to 95% (HPLC analysis). The synthesis of Mg-Al-Fe LDH is conducted according to the methodology described by Dong et al. [50], and the main chemical compositions of LDH included Al_2_O_3_ 15.37%, Na_2_O 3.34%, CaO 6.98%, MgO 35.23% and Fe_2_O_3_ 10.85%. Typically, 4 g of palygorskite (Dimaitong mine located in Linze county of Gansu Province of China) with a 200-mesh sieve was mixed with 80 mL of 1 mol/L sulfuric acid and stirred to form a homogeneous dispersion.The above dispersion was transferred into the reaction kettleafter being ultrasonically treated for 30 min, and then the mixture was hydrothermally treated at 180 °C for 12 h to collect liquid. After hydrothermal reaction, the supernatant was obtained after centrifugation and poured into a round-bottom flask, and 0.416 g of Mg(OH)_2_ was added into the flask according to the ratio of *x* = M^3+^/(M^3+^ + M^2+^) was 0.3. Then, the mixture composed of 15 g of NaOH, 1 g of Na_2_CO_3_, and 20 mL of distilled water was rapidly added into the above flask and the reaction continued for 2 h at 70 °C. After that, the solid product was centrifuged and washed with distilled water, and then hydrothermally treated at 150 °C for 12 h. Finally, the resulting solid was dried after being centrifuged and washed.

### 3.2. Preparation of Lutein/LDH Composites

The lutein/LDH compounds were prepared by a facile and green method. 100 mg of lutein was manually grinded with 1 g of synthetic LDH at room temperature for 30 min, and then the mixture was heat-treated in an oven at 60 °C for 12 h to obtain the final samples. It should be noted that the whole preparation process was carried out in dark conditions.

### 3.3. Chemical stabilityEvaluation of Lutein/LDH Composites

The chemical stability of the as-prepared samples was tested using 0.1 M NaOH, acetone, anhydrous ethanol and ethylacetate, respectively. The four mixtures obtained by dispersing 0.04 g of samples into 10 mL of the above solvents were oscillated at 150 rpm and 30 °C for 24 h, respectively. After that, the solid products were centrifuged and dried at 40 °C for measuring the chromaticity parameters.

### 3.4. Thermal Stability Evaluation of Lutein/LDH Composites

The thermal stability of the powderlutein/LDH composites were evaluated by thermogravimetric analysis (TGA) using a STA449F3 simultaneous thermal analyzer (NETZSCH-Gerätebau GmbH, Wittelsbacherstraße, German).

### 3.5. Photostability and Storage Stability of Lutein/LDH Composites

To investigate the photostability and storage stability of lutein/LDH composites, the powder samples were irradiated with visible light and heated at 55 °C, respectively. The samples were illuminated using a 4 W LED lamp (LED-1026; 50 × 255 mm, Guangdong DP Co., Ltd, Guangzhou, China) equipped with 40 LED lamp beads used as a visible light source from a distance of 10 cm at room temperature to investigate the photostability. In addition, the storage stability of the powder samples was measured by accelerated aging test. According to Equations (1) and (2) [66], the samples were heated in an oven at 55 °C for 24 h, which was equivalent to eight days at 25 °C.
(1)$$AAF=Q_{10}^{\left[\frac{T_{AA}-T_{RT}}{10}\right]}$$
(2)AAT=365/AAF
where *AAF* is accelerated aging factor, *Q*_10_ is usually 2, *TAA* represents the temperature for accelerates aging, *TRT* is the real-time aging temperature (generally 25 °C), and *AAT* is the accelerated aging time.

The maximum absorbance of the ethanol solution of lutein (2 ug/mL) appeared at 446 nm by using UV-vis spectrometer (TU-1900, PERSEE, Beijing, China) (Appendix A). Therefore, the absorbance of lutein loaded on LDH in ethanol was determined at 446 nm to investigate the light resistance and storage stability of lutein in sample. The retention index (*RI*, %) of lutein was expressed as follows [9]:(3)$$RI\left(\%\right)=\frac{A_{t}}{A_{0}}\times 100\%$$
where *A_t_* is the absorbance of lutein after being exposed to light for *t* h and *A*_0_ is the initial lutein absorbance of the samples.

### 3.6. Characterizations

The X-ray diffraction patterns (XRD) were collected on the X’pertPRO diffractometer (PANalytical Co., Almelo, The Netherlands) along with Cu-Ka radiation at 40 kV and 30 mA, the diffraction data of samples were obtained from 3 to 80° at a scanning speed of 2° per minute. The Fourier transform infrared (FTIR) spectra of samples were recorded on the range of 400–4000 cm^−1^ on a Nicolet NEXUS FTIR spectrometer (Nicolet iS50, Thermo Scientific USA, Bartlesville, OK, USA) using KBr pellets. The morphologies of the obtained samples were investigated by field emission scanning electron microscopy (FESEM, JSM-6701F, JEOL, Akisima, Tokyo, Japan). The surface area and pore volume of samples were performed on a −196 °C with N_2_ as an adsorbate using the Accelerated Surface Area and Porosimetry System (Micromeritics, ASAP2020, Atlanta, GA, USA). The zeta potentials of all samples were obtained on Malvern Zetasizer Nano system (ZEN3600, Malvern, UK)with a 633 nm He-Ne laser irradiated, in which 0.05 g of sample was dispersed into 10 mL of deionized water and sonicated for 20 min before measurement of zeta potential. Thermal gravimetric analysis (TGA) wasperformed on a STA449F3 simultaneous thermal analyzer (NETZSCH-Gerätebau GmbH, Wittelsbacherstraße, German) in the range of 30–800 °C with a heating rate of 10 °C·min^−1^ under nitrogen atmosphere. The colorimetric values and reflectance spectra of all samples were recorded on a Color-Eye automatic differential colorimeter (X-Rite, Ci 7800, Anderson & Vreeland, Inc. Corporate Headquarters, Fairfield, USA) according to the Commission Internationaledel’ Eclairage (CIE)1976 *L**, *a**, *b** colorimetric method, in which *L**, *a** and *b** represented the color lightness and the hue, respectively. The chemical composition was recorded on an E3 X-ray fluorescence spectrometer (PANalytical, Almelo, Netherlands).

## 4. Conclusions

In summary, lutein/LDH composites were successfully fabricated by loading natural lutein on Mg-Al-Fe LDH derived from spent liquor after etching palygorskite using a simple and eco-friendly grinding method. Although the relevant characterizations confirmed that lutein was mainly adsorbed on the surface of LDH, the thermal, light, and storage stability of lutein were significantly improved after incorporation of LDH.The obtained composites also exhibited good chemical resistance tolerance toward chemical agents, especially alkaline solution due to stronger chemical interactions between natural lutein and LDH. Therefore, this study provided a feasible strategy for enhancing the environmental stability of natural environmental-sensitive pigments.

## Figures and Tables

**Figure 1 molecules-25-01231-f001:**
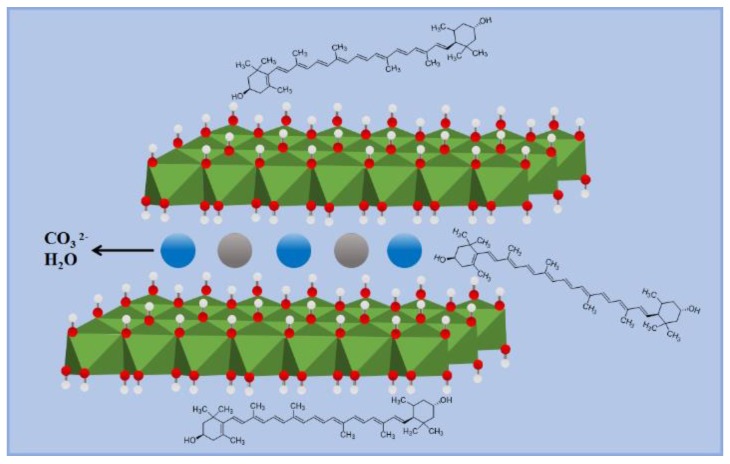
Schematic diagram of the possible interaction mechanism between lutein and layered double hydroxide (LDH).

**Figure 2 molecules-25-01231-f002:**
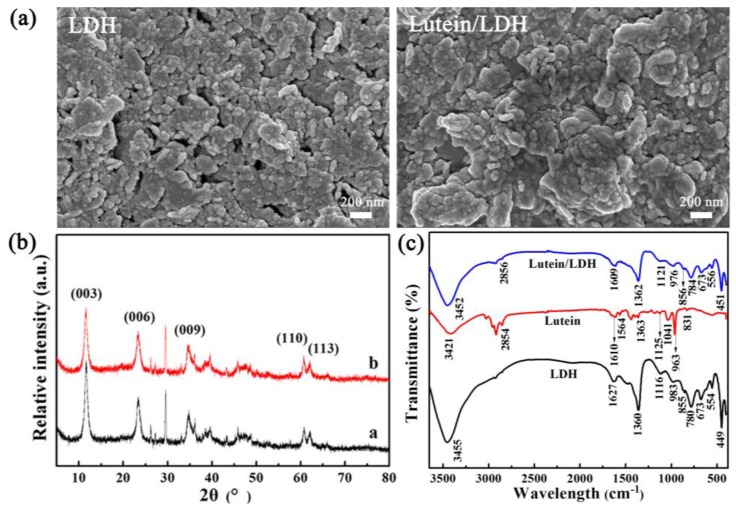
(**a**) Scanning electron microscopy (SEM) images and (**b**) X-ray diffraction (XRD) patterns of LDH and lutein/LDH composites; (**c**) Fourier Transform Infrared Spectroscopy (FTIR) spectra of lutein, LDH and lutein/LDH composites.

**Figure 3 molecules-25-01231-f003:**
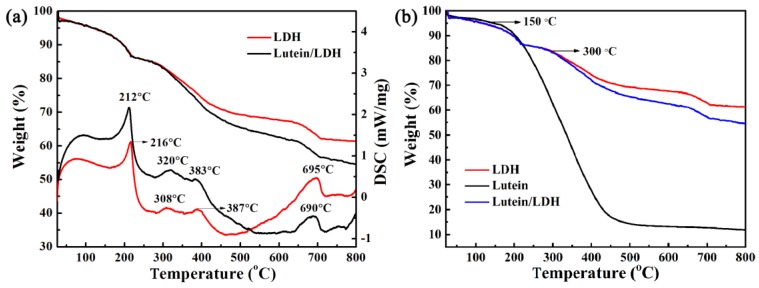
(**a**) Thermogravimetric analysis and differential scanning calorimetry (TGA-DSC) curves of LDH and lutein/LDH composites, (**b**) TGA-DSC curves of lutein, LDH and lutein/LDH hybrid composites.

**Figure 4 molecules-25-01231-f004:**
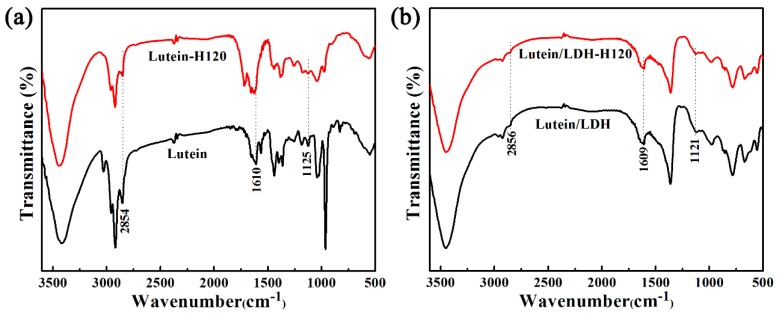
FTIR spectra of lutein and lutein/LDH before (**a**) and after (**b**) being heated at 120 °C for 4 h, respectively.

**Figure 5 molecules-25-01231-f005:**
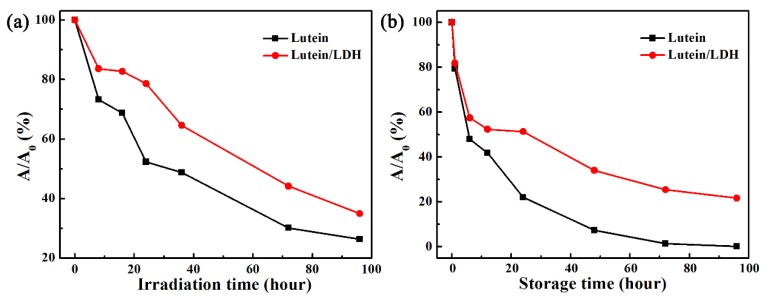
(**a**) photo-resistance and (**b**) storage stability of lutein and lutein/LDH with different exposure time.

**Table 1 molecules-25-01231-t001:** Pore structural parameters and zeta potentials of lutein, layered double hydroxide (LDH) and lutein/LDH samples.

Samples	*S_BET_* (m^2^/g)	*V_total_* (cm^3^/g)	Average Pore Width (nm)	Zeta Potentials (mV)
LDH	81.70	0.20	9.83	3.75
Lutein/LDH	62.70	0.17	10.75	−4.54

**Table 2 molecules-25-01231-t002:** Color parameters ofpure lutein, and lutein/LDH samples before and after being immersed into 0.1 M NaOH, acetone, ethanol and ethyl acetate for 24 h, respectively.

Color Parameters	Pure Lutein	Raw Samples	0.1 M NaOH	Acetone	Ethanol	Ethylacetate Acetate
*L**	44.25	55.5	51.36	50.86	50.43	52.73
*a**	37.48	16.42	17.76	15.09	15.41	14.76
*b**	43.84	44.15	42.79	30.88	29.23	31.05

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
