# Peer review of "Incorporation of Lutein on Layered Double Hydroxide for Improving the Environmental Stability"

_molecules, 2020, doi:10.3390/molecules25051231_

Round 1

Reviewer 1 Report

The paper "Incorporation of lutein on layered double hydroxide for improving the environmental stability" describes a possible route to
obtain a composite where lutein is stabilized to heat and light.
The topic of the paper is interesting but the manuscript has to be checked carefully. There many typos (there are missing spaces between words also in the title), the english could be improved in some points and the results should be described and explained in more detail. 

I have some general questions about the preparation of the material:

Was the product washed after grinding together lutein and LDH to remove the excess of lutein that could be present?

How was the % of lutein with respect to LDH chosen?

Why did the authors decide to use carbonate LDH?

I don't understand why the authors describe lutein as an anion and they mention a COO^- group. I have looked at the structure formula given by the authors and could not find one. Since the paper focuses on the interaction between LDH and lutein I think that this point is very important, so it should be clarified and the description of the interaction between lutein and LDH explained more carefully.

In the introduction I think that the description of the background should be improved. The material and the intercalation process in my opinion has to be described in more detail, citing the relevant literature.

Line 62-64: Layered double hydroxides are widely studied materials and there are already other easy and green methods that involve mechanochemical synthesis, both for the preparation of the LDH and for the intercalation, that are not mentioned in the paper. Some papers about this topic are, for example:

Inorganica Chimica Acta
Volume 470, 30 January 2018, Pages 36-50
Facile preparation methods of hydrotalcite layered materials and their structural characterization by combined techniques
DOI: 10.1016/j.ica.2017.08.007

A Simple Mechanochemical Route to Layered Double Hydroxides: Synthesis of Hydrotalcite‐Like Mg‐Al‐NO3‐LDH by Manual Grinding in a Mortar
Zeitschrift für anorganische Chemie 635(9‐10):1470 - 1475
DOI: 10.1002/zaac.200801287

ACS Omega. 2019 Nov 26; 4(22): 20072–20079.
Solvent-Mediated and Mechanochemical Methods for Anion Exchange of Carbonate from Layered Double Hydroxides Using Ammonium Salts
DOI: 10.1021/acsomega.9b03261

Line 64: The meaning of "it is scarce to synthesize LDH using metal ions derived from wastewater during clay minerals processing" is not clear to me, please rephrase this part of the sentence. 

Line 79-80 The decrease in size in my opinion is not visible in the SEM images in the manuscript.

Figure 2a: The SEM images of LDH before and after the incorporation of lutein are very similar and not very clear. 

Figure 2b and line 90-102: The XRPD patterns of the two materials are almost the same. How was the increase of d-spacing from 7.60 Å to
7.64 Å determined? It can also be an effect of "zero error" due to the sample misalignment during the measurement. A sharp peak at nearly
30° is present in both sample and it looks like an impurity. I think it should be commented. The presence of lutein is not deducible from this XRPD pattern because there is no sign of intercalation and also no sign of the presence of a crystalline or amorphous lutein phase, probably because the % of lutein is low. Maybe a longer acquisition time could improve the signal/noise ratio.

Line 94:  a COO- group that is interacting with Mg2+ ions is mentioned but in the structure formula and model of lutein reported in the manuscript (in Figure 1 and in the Supplementary information file) there isn't a COO- group
and lutein is neutral. I think the authors should clarify this point and rewrite the part about the results and the interaction between LDH and lutein.

About the intercalation of neutral molecules in layered double hydroxides I suggest to check the paper "Unusual Incorporation of Neutral
and Low Water-Soluble Guest Molecules into Layered Double Hydroxides: The Case of Cucurbit [6 and 7]uril Inclusion Hosts"
dx.doi.org/10.1021/cm102962g | Chem. Mater. 2011, 23, 1350–1352

Since the carbonate ion is very stable inside the LDH layers and lutein is a large and neutral molecule in my opinion the intercalation is very unlikely in the conditions described in the manuscript. The analysis should be more focused on explaining the nature of the interaction, also using other techniques. 

Figure 2c: in the FTIR spectra of the composite the bands attributed to lutein are barely visible.
This figure has to be improved to show more clearly the differences and the peaks attributed to lutein in the composite material and to allow the comparison between the spectra.

Paragraph 2.2.1 Chemical stability: The  explaination of the values of the colorimetric analysis  and the results should be described and explained in more detail. 
Line 140: in my opinion a few words about the reason of the better stability of layered double hydroxides in hydroxides with respect to other solvents should be written.
Line 141-142: the correlation between the color of the surnatant and the results of colorimetry on the sample should be explained better because otherwise this sentence can be confusing.

Line 117 the peaks due to the loading of luteine should be attributed.

Line 133: The negative zeta potential of lutein has to be explained and the interaction with LDH better described. How was measured the zeta potential of lutein in water since it is insoluble?  

Line 159: The increased weight loss of the LDH/lutein compound with respect to the pristine LDH demonstrate the presence of lutein in the material. I would suggest to plot also the first derivative of the TGA to highlight the weight losses.

Figure 3b: Please correct the label of the y axis, it is weight and not weight loss.

Line 181: It is not clear what the authors are meaning in the sentence where they compare their results on the composite LDH/lutein with the materials studied by Bauer et al. Moreover in the paper by Bauer et al. the dye is intercalated into LDH and not adsorbed.

In the conclusions the method of preparation of the LDH is mentioned in the materials section with the reference to literature but it is not described in this paper, therefore it should not be mentioned as part of the conclusions.
Line 264: In my opinion it is not possible to say that a structural analysis was performed and that it indicates that lutein is adsorbed. From the XRPD pattern it can only be inferred that lutein was not intercalated as it does not give any hint on the structure or the kind of interaction between lutein and LDH. 

The authors should describe in the conclusions the results obtained in the stabilization of lutein.

Reviewer 2 Report

Lutein attracts significant interests in the field of nutrition and medicinal chemistry due to its various biological activities, mainly as antioxidant, as well as for eye protection. However, its practical application is restricted due to poor solubility in water and low stability. Various formulations have been developed to improve aqueous solubility, stability and oral bioavailability of lutein, such as polymers, micelles and others. In the presented manuscript the authors demonstrate their attempts to improve the stability of lutein by its incorporation into the layered double hydroxides. However, the quality of the manuscript does not allow me to recommend it for publication in the present form. The following questions should be addressed before next submission.

1)         English should be improved with the help of native speaker.

2)         Line 29, “The carotenoid group can be classified into lutein and carotenes” is not correct. Lutein is just one example of oxygen contained carotenoids called xanthophylls.

3)  In the Introduction, lines 41-48, the authors can add more examples of increasing carotenoid’s photostability and chemical stability, as described for example in [Focsan, e.a., Supramolecular carotenoid complexes of enhanced solubility and stability – the way of bioavailability improvement. Molecules, 2019, DOI: 10.3390/molecules24213947; Apanasenko, e.a. Solubilization and stabilization of macular carotenoids by water soluble oligosaccharides and polysaccharides. Arch. Biochem. Biophys. 2015, 572, 58-65; Polyakov, e.a., Photochemical and optical properties of water-soluble xanthophyll antioxidants: aggregation vs complexation. J. Phys. Chem. B. 2013, 117, 10173-10182].

4) Lines 57, 133 and so on: why the authors called lutein and β-carotene as anionic dyes or negatively charged species? Lutein is neutral molecule.

5) Lines 92-94: “On the other hand, it was clearly established that lutein was mainly adsorbed on the surfaces of LDH due to the strong affinity of -COO- to Mg2+ located in LDH…” Where is COO- group in lutein molecule?

6) Line 141: define the L*, a* and b* parameters. The comparison with pure lutein is necessary as a control.

7) 2.2.3. Photostability and storage stability: how the experiment was performed, in solution or in solid state? In both cases the decrease of photodegradation rate might be due to light absorption or scattering in part by LDH particles. Storage stability is also not evident, there are two-modal decay kinetics, fast parts are similar in both cases, and it might be lutein decay, but slow decay might be decomposition of secondary products which absorb light at the same wavelength.

HPLC analysis will be much more reliable.

8)  Line 228: define the parameters of LED lamp.

9) 3.1. Materials: the use of Fe for synthesis LDH is not good idea, since carotenoids are very unstable in the presence of Fe ions.

10) Line 268: “this long-lasting composite is a stand out as potential alternative for food or feed supplement”. To make such conclusion the authors should present the evidence that LDH inorganic nanoparticles are absolutely nontoxic for human and animals. And to show the advantages of LDH-lutein composite over other inorganic or organic biodegradable carrier.

Author Response

Pleas see the attachment.

Round 2

Reviewer 1 Report

I think that the manuscript has improved, I suggest the authors to check all manuscript for consistency after the changes (especially where describing the interaction between lutein and LDH).

Just some minor remarks:

Line 55 "compounds consisted of the positively charged hydroxide layers, and thus the partial high-charge cations were substituted by low-charge ones while" Since LDH derive from brucite the high charge cations (trivalent) are substituting the low charge ones (divalent) so the sentence should be reversed.

Line 104 "This phenomenon might be attributed to the small size and the arrangement of lutein interlayer of LDH" This sentence is not clear, since lutein is not intercalated it doesn't form an interlayer.

Line 130: "The average pore size of composites slightly decreased from 10.75 to 9.83 cm 3 /g compared with LDH." There is confusion here... please look at the data table and pay attention to the unit of measure.

Line 235: It is not described how the grinding was performed, if manually or in a mill and, if so,  what kind of mill (e.g. ball mill, jet mill...). Since lutein has limited resistance to heat and during the grinding heat is developed due to friction forces this information is relevant.

Line 248: the form of the sample during irradiation, heating, TGA is not described...please define if it was powder / pellet / in solution

Author Response

Reviewer 1#

Comments and Suggestions for Authors

I think that the manuscript has improved, I suggest the authors to check all manuscript for consistency after the changes (especially where describing the interaction between lutein and LDH). Just some minor remarks:

Q1: Line 55 "compounds consisted of the positively charged hydroxide layers, and thus the partial high-charge cations were substituted by low-charge ones while" Since LDH derive from brucite the high charge cations (trivalent) are substituting the low charge ones (divalent) so the sentence should be reversed.

Q1 Response: Thanks for your suggestion. This sentence has been revised into “…compounds consisted of the positively charged hydroxide layers because of the partial substitution of divalent metal ions by trivalent ones, and the net positive charge on the layers was neutralized by the interlayer negative anion species and water molecules”.

Q2: Line 104 "This phenomenon might be attributed to the small size and the arrangement of lutein interlayer of LDH" This sentence is not clear, since lutein is not intercalated it doesn't form an interlayer.

Q2 Response: Thanks for your comments, and this statement has been deleted.

Q3: Line 130: "The average pore size of composites slightly decreased from 10.75 to 9.83 cm3/g compared with LDH." There is confusion here... please look at the data table and pay attention to the unit of measure.

Q3 Response: Thanks for your suggestion, and it has been revised into “The average pore size of composites slightly increased compared with that of LDH”. In addition, the lutein was removed in Table 1.

Q4: Line 235: It is not described how the grinding was performed, if manually or in a mill and, if so,  what kind of mill (e.g. ball mill, jet mill...). Since lutein has limited resistance to heat and during the grinding heat is developed due to friction forces this information is relevant.

Q4 Response: Thanks for your comments, and the relevant information has been provided in the revised manuscript.

Q5: Line 248: the form of the sample during irradiation, heating, TGA is not described...please define if it was powder / pellet / in solution

Q5 Response: Thanks for your suggestion, all samples were powder during irradiation, heating, TGA, and it also provided in the manuscript

Reviewer 2 Report

The authors have addressed all my questions. Thanks.

Author Response

Thanks for your effort for reviewing and improving this manuscript.